# Uncertainty-Based Models for Optimal Management of Energy Hubs Considering Demand Response

**Arsalan Najafi [1], Mousa Marzband [2,3,\*], Behnam Mohamadi-Ivatloo [4], Javier Contreras [5], Mahdi Pourakbari-Kasmaei [6], Matti Lehtonen [6] and Radu Godina [7]**

[1] Young Researchers and Elite Club, Sepidan Branch, Islamic Azad University, Sepidan 73611, Iran; arsalan.najafi@iausepidan.ac.ir

[2] Faculty of Engineering and Environment, Department of Maths, Physics and Electrical Engineering, Northumbria University Newcastle, Newcastle upon Tyne NE1 8ST, UK

[3] Department of Electrical Engineering, Lahijan branch, Islamic Azad University, Lahijan 44131, Iran

[4] Faculty of Electrical and Computer Engineering, University of Tabriz, Tabriz 51999, Iran; bmohammadi@tabrizu.ac.ir

[5] E.T.S. de Ingenieros Industriales, University of Castilla−La Mancha, 13071 Ciudad Real, Spain; javier.contreras@uclm.es

[6] Department of Electrical Engineering and Automation, Aalto University, Maarintie 8, 02150 Espoo, Finland; mahdi.pourakbari@aalto.fi (M.P.-K.); matti.lehtonen@aalto.fi (M.L.)

[7] UNIDEMI, Department of Mechanical and Industrial Engineering, Faculty of Science and Technology (FCT), Universidade NOVA de Lisboa, 2829-516 Caparica, Portugal; r.godina@fct.unl.pt

**\*** Correspondence: mousa.marzband@northumbria.ac.uk

**Abstract:** Energy hub (EH) is a concept that is commonly used to describe multi-carrier energy systems. New advances in the area of energy conversion and storage have resulted in the development of EHs. The efficiency and capability of power systems can be improved by using EHs. This paper proposes an Information Gap Decision Theory (IGDT)-based model for EH management, taking into account the demand response (DR). The proposed model is applied to a semi-realistic case study with large consumers within a day ahead of the scheduling time horizon. The EH has some inputs including real-time (RT) and day-ahead (DA) electricity market prices, wind turbine generation, and natural gas network data. It also has electricity and heat demands as part of the output. The management of the EH is investigated considering the uncertainty in RT electricity market prices and wind turbine generation. The decisions are robust against uncertainties using the IGDT method. DR is added to the decision-making process in order to increase the flexibility of the decisions made. The numerical results demonstrate that considering DR in the IGDT-based EH management system changes the decision-making process. The results of the IGDT and stochastic programming model have been shown for more comprehension.

**Keywords:** demand response; energy hub; information gap decision theory; stochastic programming

## 1. Introduction

### 1.1. Motivation and Problem Description

Operations of energy hub are conducted using various devices such as combined heat and power (CHP) units, electrical and thermal energy storage, boilers, power electronic devices, etc. The main issue is connecting different types of energies in the energy hub (EH) [1]. In addition, the low efficiency of plants operating with fossil fuels have encouraged researchers to use this concept for using and combining different types of energy [2]. The emergence of this concept in a restructured electricity

market can affect the decisions of consumers, operators, and generation companies. The decisions will be more complicated in the presence of uncertainties arisen by different types of technology, and therefore, it is essential to consider them in the problem to increase the robustness of the solution.

*1.2. Literature Review*

Until now, many researchers have worked in the area of energy optimization [3]. Some papers have also focused on the operation of CHP units. In Reference [4], a linear programming method is used for CHP operation with practical constraints. In Reference [5], an operation model of a CHP unit with an energy storage unit is proposed in a restructured system. Evolutionary algorithms are also utilized in References [6,7] for energy optimization. However, this research has not focused on combining multi-carrier energy systems. In Reference [1], a power flow model is suggested for EH, based on nonlinear equations describing network connections. In Reference [8], the effect of energy storage capacity and prediction horizon has been investigated on the cost optimal multi-energy supply of domestic consumers. The bi-level optimization of an EH is studied in Reference [9]. In this study, the hub operator is located in the upper level, and the consumers are in the lower level. The reliability aspects of EHs are also investigated in References [10–12]. In Reference [13], a model is proposed for an EH optimal design that intends to determine the size of the distributed energy resources, taking into account the environmental and social effects of the EH using a Benders decomposition method. Reference [14] has investigated an optimal expansion planning model for an energy hub with multiple energy systems, in which the multiple energy system planning problem would optimally distinguish the optimal investment candidates for generating units, transmission lines, natural gas furnaces, and CHPs that satisfy electricity and heat demands. The system performances associated with reliability, energy efficiency, and emission matrices is evaluated for the identified planning schedules. Residential EH in a smart home is considered in References [15–19]. A mixed integer linear programming model is presented in Reference [20] for controlling a home energy system. Some papers have investigated demand response (DR) programs on a residential scale. A residential EH with a demand side management is considered in Reference [21]. In Reference [22], a model is presented for a residential EH in a smart home considering a heat pump water heater and the coordination of sources and carbon emissions. Reference [23] investigates storage systems for improving the technical and financial aspects of a residential renewable-based EH. In Reference [24], an algorithm is suggested for a micro-EH in a grid with distributed generators. A bidding strategy is proposed in References [25,26], taking into account both day-ahead (DA) and real-time (RT) electricity markets using a three-stage stochastic programming method. The maintenance problem of an EH is investigated in Reference [27], considering the water energy as an output of the EH besides the electrical and thermal energies. The maintenance problem is formulated and solved with respect to its reliability indices. A game theory model is used for multiple EHs in Reference [28] for their economic scheduling. The authors in Reference [29] have studied DR for the thermal demand of an EH.

The Information Gap Decision Theory (IGDT) method has been used frequently in the robust optimization of power systems. For example, in References [30–35], a bidding/offering strategy is presented for purchasing and selling power to manage the electricity price forecasting errors using IGDT. The bidding/offering strategy is also studied in smart grids in Reference [36]. A bidding strategy is suggested in Reference [37], considering DR for covering the electricity of large utilities. The offering price of wind power producers is investigated in References [38,39]. A multistage transmission expansion planning under wind power uncertainties is obtained using the IGDT method in References [40–43]. A security constrained unit commitment (SCUC) with undispatchable resources with a high penetration of wind power for retaining the load-generation balance is proposed in References [44,45], and the frequency management is presented in References [46,47].

*1.3. Paper Contributions*

In this paper, a new model is proposed for EH operation based on the information gap decision theory (IGDT) considering DR. IGDT is used in order to show the robustness of the model proposed against electricity prices and wind uncertainties. IDGT is a non-probabilistic optimization method in which the decisions are made without any assumptions for the probability of the uncertainties. This is useful when a wide range of uncertainties exists [48,49]. Unlike other uncertainty modeling approaches, the objective of the IGDT method is to maximize the tolerable horizon of uncertainty while satisfying a predetermined objective [38]. Thus, the operators or owners of EHs may make decisions to distinguish the critical decisions in the presence of a high level of uncertainties.

The IGDT model was applied on energy hubs before but without considering the electricity price uncertainty [50]. However, the IGDT is extended by considering the electricity price uncertainty in the present work. A time of use (TOU) DR program is also considered in the problem in order to give more flexibility to large consumers. It can be noted that the variability of gas prices versus electricity prices is very low. Therefore, the gas price is considered fixed. Consequently, it is not required to take DR into account for heat demands. The reason behind this is that the price of procuring heat is the same at all hours. Briefly, the contributions of the paper are as follows:

- Proposing a mixed integer nonlinear (MINLP) model for an EH based on a robust model.
- Proposing an IGDT model considering price and wind uncertainties.
- Proposing DR with IGDT simultaneously in the EH.
- Considering an energy storage system (ESS) in the IGDT-based problem.

## 2. Robust Model of an Energy Hub

An EH can be small scale, e.g., a residential hub, or large scale, such as a whole country. In this paper, an EH is considered to describe the decisions of large industrial consumers or even part of a city demand. Figure 1 shows the EH under study, which includes the electricity market, wind turbine, and natural gas data as inputs to satisfy electricity and heat demands. The proposed hub has a boiler, a CHP, and an ESS. The CHP and the boiler are both fed by natural gas. Natural gas prices have a low variability in the short-term compared to electricity prices. Therefore, gas prices are assumed to be fixed. The owner of the hub purchases or generates power from three sources: DA and RT electricity markets and a wind turbine. For a more accurate management of the procured power (specifically in peak hours), an ESS is utilized. The ESS can be charged from the RT electricity market or from a wind turbine. The CHP output has not been connected to the ESS for charging due to its fixed feeding price. In other words, the CHP can generate electricity at a fixed tariff at all hours, and it is not essential to connect it to the ESS. On the other hand, the CHP can sell power to the RT electricity market in order to obtain revenue. Heat is also generated by the boiler and the CHP unit.

In a classic optimization problem, the hub owner tries to minimize the total cost of procuring electric power including the costs of the electrical and natural gas networks in the presence of uncertainties. However, sometimes the owners (or operators) may like to increase the system's robustness against the uncertainties. By doing so, they can make decisions against the worst conditions related to the uncertainties. Here, the owner makes a robust decision in the presence of wind and RT electricity price uncertainties. In addition, a large consumer can utilize the time of use demand response (TOU-DR) for electricity demand during the decision-making process. The conditions of the input sources including uncertainties or variability are shown in Figure 1.

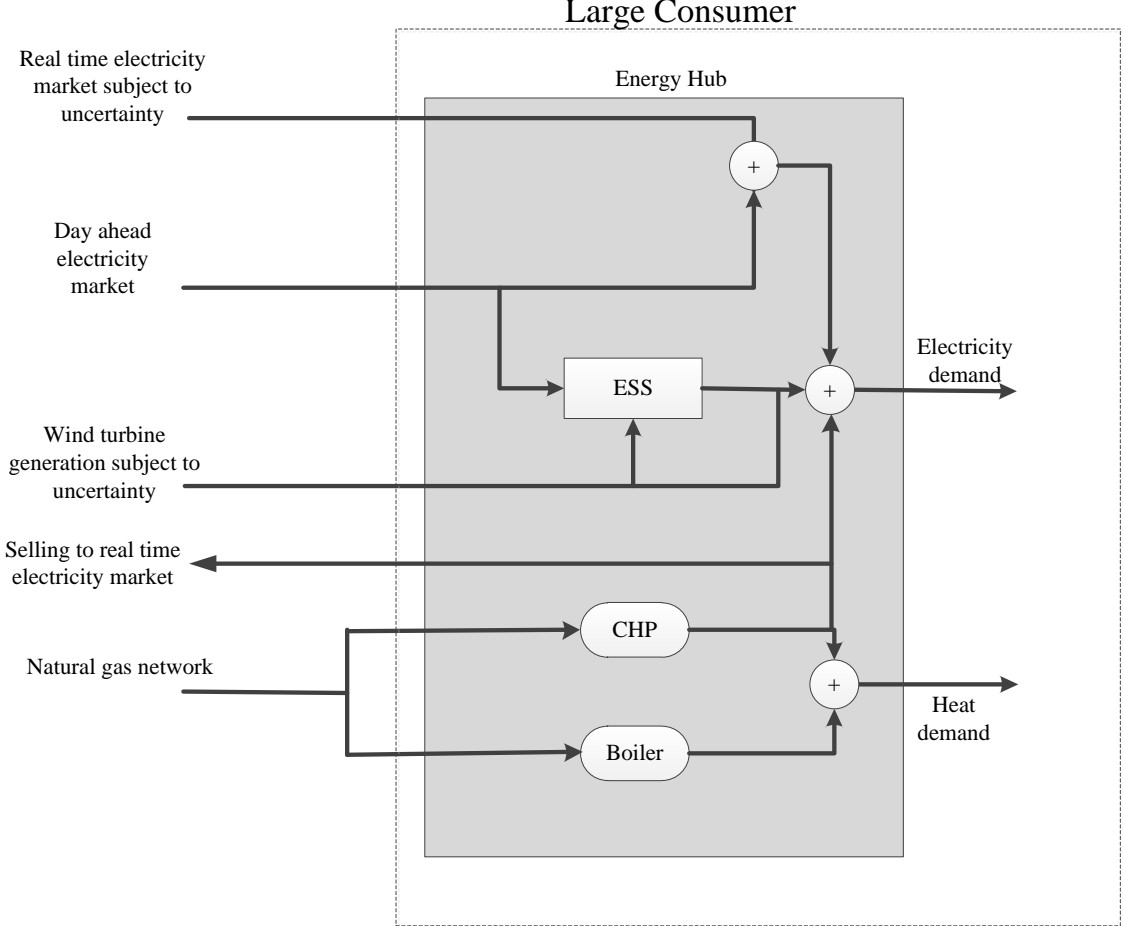

**Figure 1.** The energy hub under study.

## 3. Problem Modeling

### 3.1. CHP and Boiler Models

A CHP uses gas and generates power and heat. The power and heat generated by a CHP are mutually dependent. This dependency is considered using a feasible operation region as shown in Figure 2. The electrical and thermal outputs of the CHP must be positioned within the enclosed area formed by the points *A*, *B*, *C*, and *D*. Interested readers are referred to Reference [51,52]. The cost of CHP is related to both the heat and electricity generated. Actually, it is quadratic due to the thermodynamic relation of heat and electricity. The fuel cost characteristics of the CHP unit are as follows [53]:

$$C_{\text{CHP}}(t) = \sum_{i=1}^{I} a_i (P_i^{\text{CHP}})^2(t) + b_i P_i^{\text{CHP}} + c_i + d_i (H_i^{\text{CHP}})^2(t) + e_i H_i^{\text{CHP}} + f_i P_i^{\text{CHP}} H_i^{\text{CHP}} \tag{1}$$

where $P_i^{\text{CHP}}(t)$ and $H_i^{\text{CHP}}(t)$ are the power and heat generated by the *i*th CHP, respectively; the parameters $\eta_e^{\text{CHP}}$ and $\eta_h^{\text{CHP}}$ are the conversion coefficients from gas to power and heat; and $a_i$, $b_i$, $c_i$, $d_i$, $e_i$, and $f_i$ are the cost coefficients of the CHP.

The dependency between the generated power and the heat is not considered here for the sake of simplicity [54].

The boiler is also fed by gas and generates heat. The total cost of the boiler unit is supposed to be a linear function as follows [54]:

$$C_{\text{Blr}}(t) = \sum_{j=1}^{J} h_j H_j^{\text{Blr}} \tag{2}$$

where $H_j^{\text{Blr}}$ and $h_j$ are the heat generated by the boiler and the cost function coefficient of the boiler unit, respectively. The heat generated by the boiler is bounded as follows:

$$0 \leq H_j^{\text{Blr}}(t) \leq H_{j,\text{max}}^{\text{Blr}} \tag{3}$$

where $H_{j,\text{max}}^{\text{Blr}}$ is the maximum generation of the boiler.

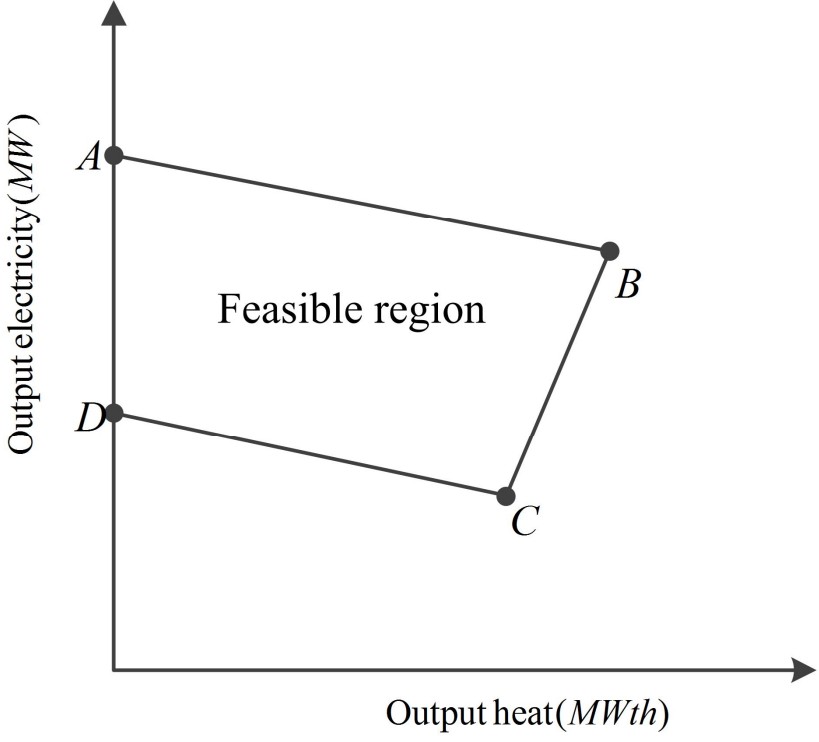

**Figure 2.** The feasible operation region of combined heat and power (CHP) [51].

### 3.2. RT and DA Electricity Price Models

In this paper, DA and RT electricity markets are taken into account. The RT market is closer to the consumption time. Hence, uncertainty is considered only in the RT market, since the DA market does not have uncertainty in the model. The realization of the DA electricity market occurs before the RT market, and it is assumed that the decisions are made when the DA price realization is completed. Therefore, the DA electricity price is assumed deterministic, and uncertainty is only considered in RT prices. In reality, RT decisions are made every ten minutes, but in our case study, it is assumed that they are made hourly for the sake of simplicity. The total cost of the RT and DA markets are described mathematically as follows:

$$C^{\text{DA}} = \sum_{t=1}^{T} P^{\text{DA}}(t).\lambda^{\text{DA}}(t) \tag{4}$$

$$C^{\text{rt}} = \sum_{t=1}^{T} P^{\text{RT}}(t).\bar{\lambda}^{\text{RT}}(t) \tag{5}$$

where $C^{\text{rt}}$, $C^{\text{DA}}$, $P^{\text{RT}}(t)$, $P^{\text{DA}}(t)$, $\overline{\lambda}^{\text{RT}}(t)$, and $\lambda^{\text{DA}}(t)$ are the total cost of purchasing from the RT market, the total cost when facing the DA market, the energy purchased from the RT market, the energy purchased from the DA market, the expected price of the RT market, and the DA electricity price, respectively.

### 3.3. Energy Storage Modeling

Energy storage should be modeled considering its charging/discharging modes. Here, the ESS is modeled as follows [11]:

$$\text{SOC}(t) = \text{SOC}(t-1) + (P^{\text{ch}})\eta^{\text{ST}}_{\text{ch}} - \frac{P^{\text{dch}}(t)}{\eta^{\text{ST}}_{\text{Dch}}} \tag{6}$$

$$\text{SOC}(t_0) = \text{SOC}(t_T) \tag{7}$$

$$E^{\text{st}}_{\text{min}} \leq \text{SOC}(t) \leq E^{\text{st}}_{\text{max}} \tag{8}$$

$$P^{\text{dch}}(t) \leq E^{\text{Dch}}_{\text{max}} \cdot \zeta_{\text{dch}} \tag{9}$$

$$P^{\text{ch}}(t) \leq E^{\text{st}}_{\text{max}} \cdot \zeta_{\text{ch}} \tag{10}$$

$$\zeta_{\text{ch}} + \zeta_{\text{dch}} \leq 1 \tag{11}$$

where $\text{SOC}(t)$ is the state of charge at time $t$; $\eta^{\text{ST}}_{\text{ch}}$ and $\eta^{\text{ST}}_{\text{Dch}}$ are the charging and discharging efficiencies, respectively; $P^{\text{ch}}$ and $P^{\text{dch}}$ are the charged and discharged power, respectively; the parameters $E^{\text{st}}_{\text{min}}$ and $E^{\text{st}}_{\text{max}}$ are the minimum and maximum capacities of the ESS; and $\zeta^{\text{ST}}_{\text{ch}}$ and $\zeta^{\text{ST}}_{\text{Dch}}$ are binary variables to show the charge/discharge state of battery. It can be seen in Equation (6) that the SOC in each hour is obtained by considering the charging and discharging amounts of the ESS in that hour. Based on Equation (7), the SOC at the beginning and the end must be equal. The SOC is limited by its bound in Equation (8). The amount of charging/discharging is limited to the maximum capacity of the ESS in Equations (9) and (10). According to Equation (11), the charging and discharging modes cannot occur simultaneously.

### 3.4. Wind Energy Modeling

Wind generation is a function of wind speed. The function is described based on operational characteristics as follows [55]:

$$P^W(t) = \begin{cases} 0 & 0 \leq S(t) \leq V_{\text{cin}} \\ A + B \times S^2(t) & V_{\text{cin}} \leq S(t) \leq V_n \\ P_n & V_n \leq S(t) \leq V_{\text{cout}} \\ 0 & V_{\text{cout}} \leq S(t) \end{cases} \tag{12}$$

where $V_n$ is the nominal wind speed and $V_{\text{cin}}$ and $V_{\text{cout}}$ are the cut-in and cut-out wind speeds, respectively. Electricity generation starts when the speed reaches $V_{\text{cin}}$ and the generation continues up to $V_{\text{cout}}$. Nominal generation starts when it reaches $V_n$; $P^W(t)$ is the generated power, and $S_t$ is the wind speed. The constant coefficients $A$, $B$, and $C$ are obtained from $V_n$, $V_{\text{cin}}$, and $V_{\text{cout}}$. More information can be found in Reference [56].

### 3.5. Demand Response Model

In a smart environment, consumers can use DR programs for flexible load management. By applying a DR program, consumers can adjust their consumption with regard to the volatility of the electricity pool market with the aim of minimizing the total energy cost. They can decrease their consumption in peak hours and increase their consumption in off-peak hours instead. This means that a shifting occurs only in energy consumption. The time of use demand response (TOU-DR) is

used here in order to give more flexibility to the large consumers for tuning their consumption and decreasing their total cost. Consumers can change their energy use from peak hours to off-peak hours based on the usage time of DR [57,58]. In TOU-DR, load shifting occurs and the decreased loads are supplied in other periods. Therefore, the load values decrease in some periods and increase in other periods. Equations (13)–(18) show the DR model as follows:

$$D_e(t) = \mathrm{ldr}(t) + (1 - \mathrm{DR}(t)) \times D_e^0(t) \tag{13}$$

$$\sum_{t=1}^{T} \mathrm{ldr}(t) = \sum_{t=1}^{T} \mathrm{DR}(t).D_e^0(t) \tag{14}$$

$$\mathrm{ldr}(t) \leq \mathrm{inc}(t).D_e^0(t) \tag{15}$$

$$\mathrm{DR}(t) \leq \mathrm{DR}^{\max}(t).u_{\mathrm{dr}}(t) \tag{16}$$

$$\mathrm{inc}(t) \leq \mathrm{inc}^{\max}.u_{\mathrm{inc}}(t) \tag{17}$$

$$u_{\mathrm{dr}}(t) + u_{\mathrm{inc}}(t) \leq 1 \tag{18}$$

Equation (13) shows the amount of load after applying the DR program. According to Equation (14), the decreasing total amount of load should be equal to the increasing value. The increasing amount of load should be a part of the base load, according to Equation (15). The amount of decrease/increase should be lower than a specific percentage (see Equations (16) and (17)). Equation (18) dictates that an increasing or decreasing load cannot occur simultaneously.

## 4. Information Gap Decision Theory

In this paper, IGDT is used to model the robustness of the problem against uncertainties associated with wind generation and electricity market prices. IGDT is a non-probabilistic method and does not need further information about uncertainties like probability density functions, fuzzy membership functions, or time series. Since it does not need to make any assumptions about the probability distribution of uncertain variables, it is useful when a high level of uncertainty exists in the problem [59]. Uncertainties may lead the problem to having more cost/less profit or less cost/more profit. IGDT addresses these two conflicting issues using two immunity functions, i.e., robustness and opportunity [30,60].

The three pieces that are needed in the IGDT problem are an uncertainty model, a robustness function, and an opportunity function. A robust IGDT model is presented here while the opportunity function is out of scope of this paper. Uncertainty in the IGDT model is described as follows [38]:

$$U(\alpha, \bar{\rho}) = \left\{ \rho : \left| \frac{(\rho - \bar{\rho})}{\bar{\rho}} \right| \leq \alpha \right\}, \quad \alpha \geq 0 \tag{19}$$

where $\bar{\rho}$ is the expected value of the uncertain parameters and $\rho$ and $\alpha$ represent the ranges of uncertainty. Equation (19) denotes the uncertainty bounds as shown in Figure 3.

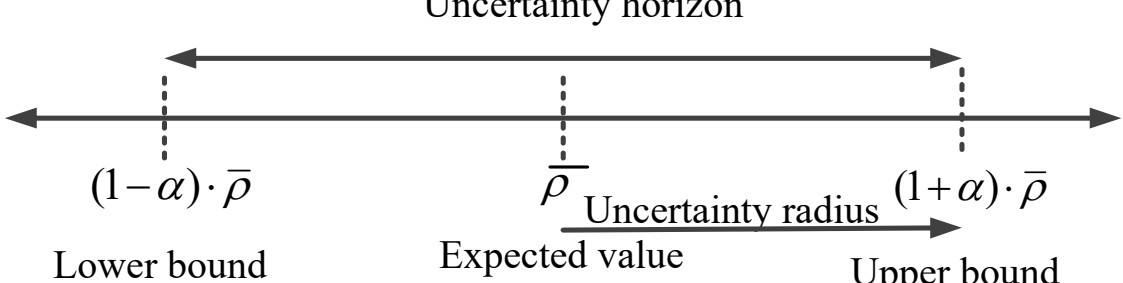

**Figure 3.** The uncertainty bounds [38].

This strategy is used when the uncertainty has unfavorable effects in the objective function. In the robustness function, the maximum uncertainty radius ($\alpha$) is determined by dedicating a specific risk value to the base deterministic mode of the objective function. In other words, the objective function is robust against uncertain input parameters, and the amount of objective function is lower than the specific value within the obtained radius of uncertainty. These decisions are made by risk-averse decision makers. The robustness function is described mathematically as follows:

$$\max_{x} \ \alpha \tag{20}$$

$$H_i(x,\overline{\rho}) \leq 0 \tag{21}$$

$$G_j(x,\overline{\rho}) = 0 \tag{22}$$

$$\Delta = (1+\beta).f_b(x,\overline{\rho}) \tag{23}$$

$$f_b(x,\overline{\rho}) \leq \Delta \tag{24}$$

$$(1-\alpha).\overline{\rho} \leq \rho \leq (1+\alpha).\overline{\rho} \tag{25}$$

$$0 \leq \beta \leq 1 \tag{26}$$

where $G_j(x,\overline{\rho})$ and $H_j(x,\overline{\rho})$ are sets of equality and inequality constraints, respectively; $\Delta$ is the maximum amount of allowable increase in the base cost determined by the decision maker; $f_b(x,\rho)$ is the base cost; and $\beta$ is the risk level.

## 5. Problem Formulation

It is necessary to propose the deterministic formulation of the EH management before the robust formulation. This is because a base cost is required to solve the robust problem, which is obtained by solving the problem without considering uncertainty. Therefore, the problem formulation is presented in two parts.

### 5.1. Energy Hub Management Formulation without Uncertainty

The expected values of the uncertain parameters should be considered in order to model the problem without uncertainties. Therefore, the expected values of wind generation and RT electricity prices should be present in the model. The deterministic model is as follows:

$$f_b = \mathbf{min} \sum_{t=1}^{T} \left[ P^{\mathrm{DA}}(t)\lambda^{\mathrm{DA}}(t) + P^{\mathrm{RT}}(t)\overline{\lambda}^{\mathrm{RT}}(t) + C_{\mathrm{CHP}}(t) + C_{\mathrm{Blr}}(t) - P^{\mathrm{CHP,M}}(t)\overline{\lambda}^{\mathrm{RT}}(t) \right] \tag{27}$$

$$0 \leq H_j^{\mathrm{Blr}}(t) \leq H_j^{\mathrm{max,Blr}} \tag{28}$$

$$D_e(t) = P^P(t) + P_{\mathrm{LD}}^{\mathrm{rt}} + \sum_{i=1}^{I} P_i^{\mathrm{CHP,L}}(t) + \overline{P}_{\mathrm{LD}}^{\mathrm{WT}}(t) + P^{\mathrm{dch}}(t) \tag{29}$$

$$D_h(t) = \sum_{j=1}^{J} H_j^{\mathrm{Blr}} + \sum_{i=1}^{I} H_i^{\mathrm{CHP}}(t) \tag{30}$$

$$\mathrm{SOC} = \mathrm{SOC}(t-1) + (P_{\mathrm{ST}}^{\mathrm{rt}}(t) + \overline{P}_{\mathrm{ST}}^{\mathrm{WT}}(t))\eta_{\mathrm{Ch}}^{\mathrm{ST}} - \frac{P^{\mathrm{dch}}(t)}{\eta_{\mathrm{Dch}}^{\mathrm{ST}}} \tag{31}$$

$$\mathrm{SOC}(t_0) = \mathrm{SOC}(t_T) \tag{32}$$

$$E_{\min}^{\mathrm{st}} \leq \mathrm{SOC}(t) \leq E_{\max}^{\mathrm{st}} \tag{33}$$

$$P^{\mathrm{dch}}(t) \leq E_{\max}^{\mathrm{Dch}}.\zeta_{\mathrm{dch}} \tag{34}$$

$$P^{\text{ch}}(t) \leq E_{\text{max}}^{\text{st}}.\zeta_{\text{ch}} \tag{35}$$

$$\zeta_{\text{ch}} + \zeta_{\text{dch}} \leq 1 \tag{36}$$

$$\overline{P}^{\text{WT}} = \overline{P}_{\text{ST}}^{\text{WT}} + \overline{P}_{\text{LD}}^{\text{WT}} \tag{37}$$

$$P^{\text{rt}}(t) = \overline{P}_{\text{ST}}^{\text{rt}} + \overline{P}_{\text{LD}}^{\text{rt}} \tag{38}$$

$$P^{\text{CHP}}(t) = P^{\text{CHP,M}}(t) + P^{\text{CHP,L}}(t) \tag{39}$$

and Equations (1) and (2).

Equation (27) shows that the total cost of the energy carriers is equal to the power purchased from the DA market, from the RT market, and from the natural gas network. Equation (28) represents the boiler generation bounds. In Equation (29), the electricity demand balance is presented. The load is fed by the DA and RT markets, discharging energy of the ESS, the wind turbine, and the CHP unit. Parameters $\overline{P}_{\text{LD}}^{\text{rt}}$ and $\overline{P}_{\text{LD}}^{\text{WT}}$ show the contributions of RT purchases and wind unit generation, which are transferred directly to the load. Similarly, Equation (30) shows the heat balancing constraint, including the boiler and CHP unit contributions. The ESS constraints are presented in Equations (31)–(36). The RT electricity market and wind unit generation are used for charging the ESS.

The parameters $\overline{P}_{\text{ST}}^{\text{rt}}$ and $\overline{P}_{\text{ST}}^{\text{WT}}$ show the contributions of the purchased electricity and the wind generation used for charging the ESS. Equations (37) and (38) show that the total wind generation and electricity purchased from the RT market are divided into two parts. The first part is directly transferred to the load, and the second part is used for charging the ESS. The CHP may also transfer power to the load or sell it to the RT market as shown in Equation (39).

## 5.2. Robust Energy Hub Management Formulation

In robust management, the uncertain parameters are placed within their intervals. As mentioned, in this paper, the uncertainties are electricity market price and wind turbine generation. Finally, the robust model is obtained as follows:

$$\textbf{min } -\alpha \tag{40}$$

$$(1-\alpha)\overline{\lambda}^{\text{RT}}(t) \leq \lambda^{\text{RT}}(t) \leq (1+\alpha)\overline{\lambda}^{\text{RT}}(t) \tag{41}$$

$$(1-\alpha)\overline{P}^{\text{WT}}(t) \leq P^{\text{WT}}(t) \leq (1+\alpha)\overline{P}^{\text{WT}}(t) \tag{42}$$

$$\max\{\Delta \leq (1+\beta)f_b\} \tag{43}$$

$$\Delta = \sum_{t=1}^{T} \left[ P^{\text{DA}}(t)\lambda^{\text{DA}}(t) + P^{\text{RT}}(t)\lambda^{\text{RT}}(t) + C_{\text{CHP}}(t) + C_{\text{Blr}}(t) - P_{\text{CHP,M}}(t)\lambda^{\text{RT}}(t) \right] \tag{44}$$

$$D_e(t) = P^{P}(t) + P_{\text{LD}}^{\text{rt}}(t) + \sum_{i=1}^{I} P_i^{\text{CHP,L}}(t) + P_{\text{LD}}^{\text{WT}}(t) + P^{\text{dch}}(t) \tag{45}$$

$$\text{SOC}(t) = \text{SOC}(t-1) + (P_{\text{ST}}^{\text{rt}}(t) + P_{\text{ST}}^{\text{WT}}(t))\eta_{\text{Ch}}^{\text{ST}} - P^{\text{dch}}(t)\eta_{\text{Dch}}^{\text{ST}} \tag{46}$$

$$P^{\text{WT}} = P_{\text{ST}}^{\text{WT}} + P_{\text{LD}}^{\text{WT}} \tag{47}$$

and Equations (1), (2), (13)–(17), (30) and (32)–(39).

It can be seen in Equation (40) that the uncertainty radius should be maximized. It is considered negative for making it a min-max problem. Equations (41) and (42) denote the bounds of uncertainty for the uncertain parameters. The total cost should be lower than a specific value based on Equation (43). The critical cost is shown in Equation (44).

Therefore, it can be said that the worst case of uncertainties for electricity prices and wind generation occurs in $(1+\alpha)\overline{\lambda}^{\text{RT}}$ and $(1-\alpha)\overline{P}^{\text{WT}}$.

## 6. Numerical Results

### 6.1. Case Study

The aforementioned EH in Figure 1 is used to show the effectiveness of the proposed model of EH robust and stochastic management considering DR. As mentioned, electricity energy is procured in various ways including purchasing power from the DA and RT electricity markets, generating power by using wind turbine and the CHP units, and using the ESS. The CHP can also sell power to the RT market. Furthermore, heat is procured in two ways, from the CHP and the boiler units. Market prices and electricity demands are obtained from the New York ISO [61]. Figure 4 shows the electricity and heat demands, and Figure 5 depicts the RT and DA electricity prices. The RT decisions are made every ten minutes in reality, but the decisions of the RT market are set for an hour in this paper. This is customary for the sake of simplicity and considering the overlapping hours with the DA market. Since the DA decisions are made each hour, the reactions of the RT and DA markets are also available each hour. The electricity load data is about 5% of the total load of New York City. The CHP feasible operation region is enclosed by the coordinates $A(0, 24.7)$, $B(18, 21.5)$, $C(10.48, 8.1)$, and $D(0, 9.88)$ [54]. The heat demand data is taken from the HOMER software data sheet [62]. The wind speed data is given in Reference [63]. Note that all data periods are 24 hours and that a summary of the data characteristics is provided in Table 1. It is assumed that the boiler output is large enough to supply the heat demand on its own. The maximum capacity of the ESS is considered to be 55 MWh. The charging/discharging efficiencies are also considered to be 0.9 [11]. The nominal generation of the wind turbine is 20 MW, and its characteristics are given in Reference [64]. The value of the risk level ($\beta$) is 0.1. The EH management problem is formulated as an MINLP problem, and it is solved using the SBB solver in the GAMS software environment [65].

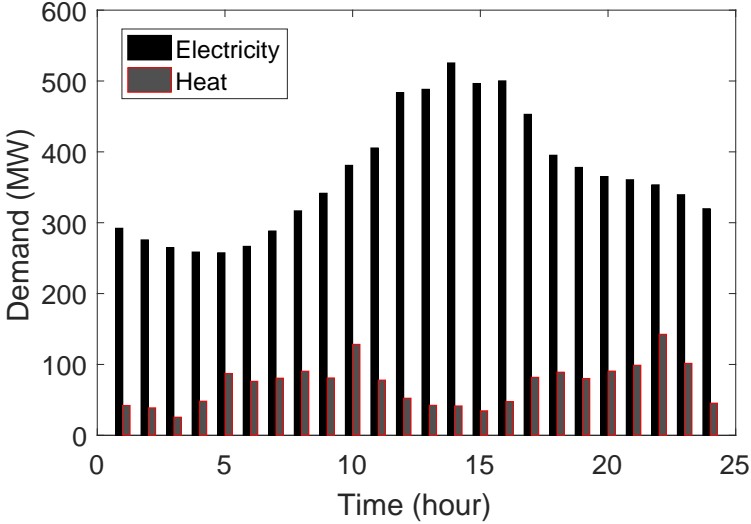

**Figure 4.** Power and heat demands.

**Table 1.** The characteristic data of the case study.

| Device (Coefficient) | Value |
|---|---|
| BSS maximum capacity | 55 MWh |
| Risk coefficient | 0.1 |
| Charging/discharging efficiencies | 0.9 |
| CHP FOR points | *A(0, 24.7)*, *B(18, 21.5)*, *C(10.48, 8.1)*, and *D(0, 9.88)* |
| Wind power maximum output | 20 MW |

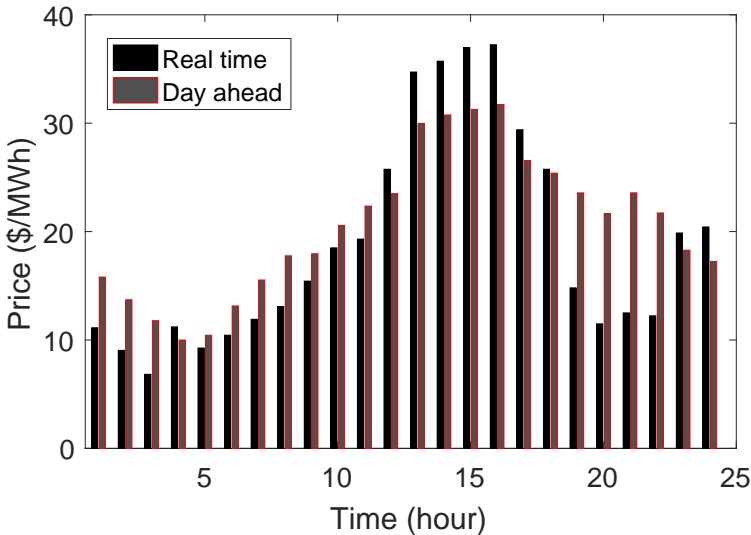

**Figure 5.** Real-time and day-ahead electricity prices.

*6.2. Simulations and Discussion*

The simulations are performed for the aforementioned EH with 0, 10, 20, and 30% of DR. The effect of the DR percentage on the base load is presented in Figure 6. It is found that DR is directly proportional to the load during off-peak hours and inversely proportional to the load in peak hours.

Figure 7 depicts the amount of power purchased from the DA electricity market. It can be seen that less power is purchased during peak hours by increasing the percentage of DR. This is due to a decrease in the amount of electricity demand in the peak hours by increasing DR. As a result of the increase in DR, the purchase of power during off-peak hours also increases. This is caused by the increase in load during the off-peak hours in the DR program. Figure 8 shows the total RT power purchased and stored by the ESS. Power is purchased from the RT market during the hours 1 to 4 and 18 to 23. Regarding the DA and RT market prices, it can be seen that RT prices are lower than DA prices in the mentioned hourly ranges and that the opposite occurs in the other hours. Therefore, no power is purchased in the other hours. In addition, the ESS is charged by the RT market power when the RT prices are lower. On the other hand, the higher the amount of DR, the higher the amount of power purchased. This is due to the rising demands in the off-peak hours. Additionally, a high level of DR is related to a lower uncertainty radius and, consequently, to lower RT prices and higher wind turbine generation. This issue is shown in Figure 9 for the RT prices. The amount of power sold to the RT market and transferred to the load by the CHP has been depicted in Figure 10. The selling price is equal to the RT price. Hence, higher RT prices result in more sales to the market by the CHP. In the other hours, the CHP helps to meet the electricity demand. DR can not affect the CHP, and minimum changes occur in its generation by changing the DR percentage. This happens because the CHP unit is a low cost unit that can also sell power and because its cost has no conflict with the economic goals of the DR program. The heat generated by the boiler and the CHP is shown in Figure 11. The maximum heat generated by the CHP is between 8 and 20 MW, when the CHP sells power to the market. It is clear that the remaining required heat has to be provided by the boiler as shown.

The charging/discharging states of the ESS are indicated in Figure 12. More charging states occur in the off-peak hours, when the RT electricity price is low. Discharge occurs in the peak hours (i.e., hours 14, 15, and 16). According to the RT prices, discharging occurs at the beginning of the period and, then, charging starts when the prices go down. In addition, a stochastic programming model (a well-known model for handling the uncertainty problem) has been conducted for a better understanding of the problem (see Table 2). Five scenarios from each uncertain source have been generated using a normal distribution function with a 20% standard deviation. The mean values of the these sources are assumed to be equal to the forecasted values which had been used in implementation

of IGDT. Then, 25 scenarios have been generated after the combination of scenarios. Interested readers can refer to Reference [66] for more details about the scenario generation.

The given expected cost results from using the forecasted values of the uncertain parameters. The total cost obtained by the stochastic programming method is lower than both the expected cost and the critical cost. It can also be observed that a higher DR percentage results in a smaller uncertainty radius and a lower cost because, by increasing the percentage of DR, the electricity demand in the peak hours decreases. By reducing the peak load, there is a similarity between the problem and a deterministic one, and consequently, the uncertainty radius decreases. By raising the percentage of DR, the expected cost is reduced due to a decrease in the demand during peak hours.

A sensitivity analysis was performed on the risk coefficient and DR, and the results are provided in Table 3. The risk parameter was changed from 0.02 to 0.12 with a 0.02 step with respect to the two states of DR (0 and 30% DR). The five outputs of the problem include the uncertainty radius, the total power purchased from the RT and DA markets, and the total power and heat generated through the CHP in all hours. A higher value of the risk parameter results in a lower value of the uncertainty radius, meaning that the uncertain parameters can vary in a wider interval when the risk level is high. A reverse relation can be seen between risk and RT market participation. By raising the risk coefficient, the robustness of the problem increases. Therefore, it is natural to observe a decrease in the power produced by uncertain sources such as RT market participation and, on the other hand, to have an increase in deterministic sources such as DA market participation. However, the CHP is a deterministic power source, but its generation has a direct relation with the risk coefficient due to the CHP role in the problem in which it can sell power to the RT market. By raising the DR percentage, the participation in the RT market increases, and on the other hand, the power purchased from the DA market decreases because DR shifts the peak demand to off-peak hours. Therefore, by comparing the RT and DA market prices in the peak and off-peak hours, it can be said that the RT and DA markets are complementary and that their sum in each state of DR is close. In the case of CHP generation, a reduction occurs considering 30% of DR. As aforementioned, the DR shifts the peak hour demand to off-peak hours. In the first state of DR (0% of DR), plenty of peak hours and peak prices exist. Thus, the CHP as a low-cost source is required, while, in the second state (30% of DR), by shifting the demand from the peak hours to the off-peak hours, the need for the CHP is reduced. In other words, it cannot be said that the CHP is a low-cost source compared to the off-peak prices.

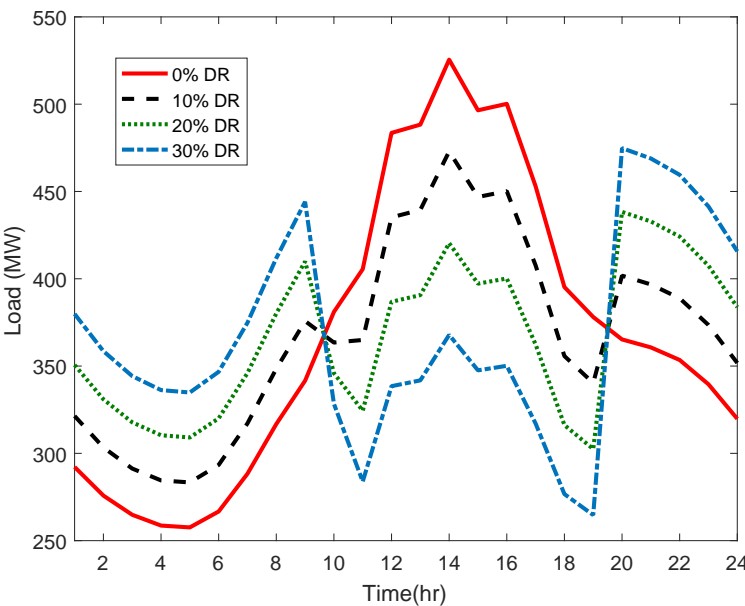

**Figure 6.** The ower demand after the demand response (DR) program.

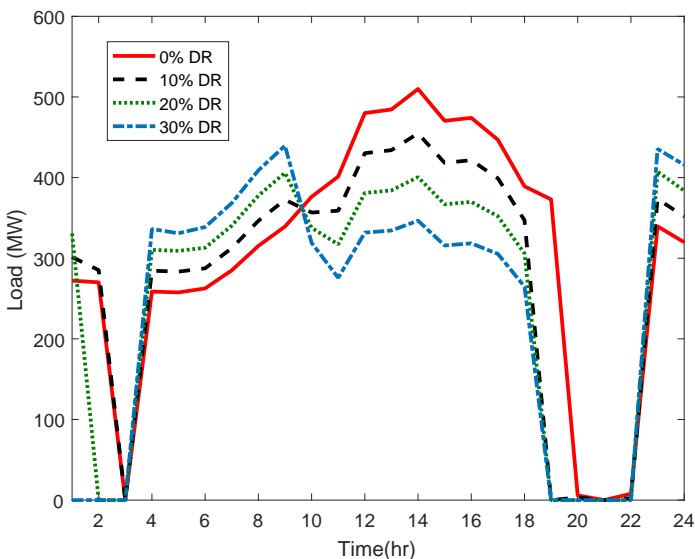

**Figure 7.** The power purchased from the day-ahead electricity market.

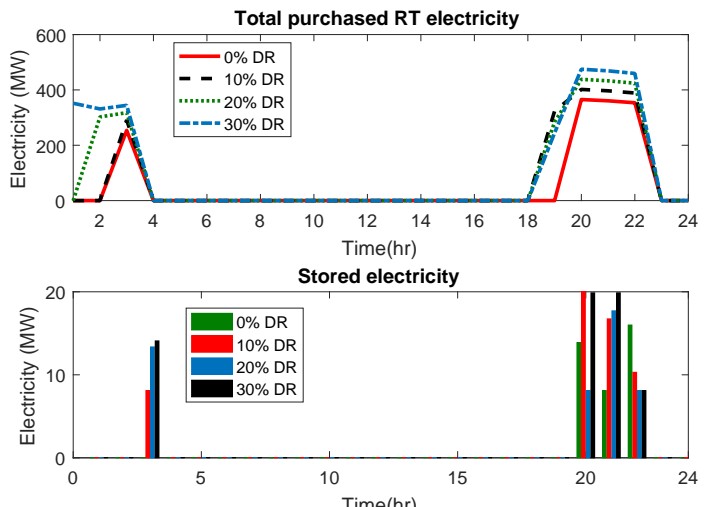

**Figure 8.** The power purchased from the real-time electricity market.

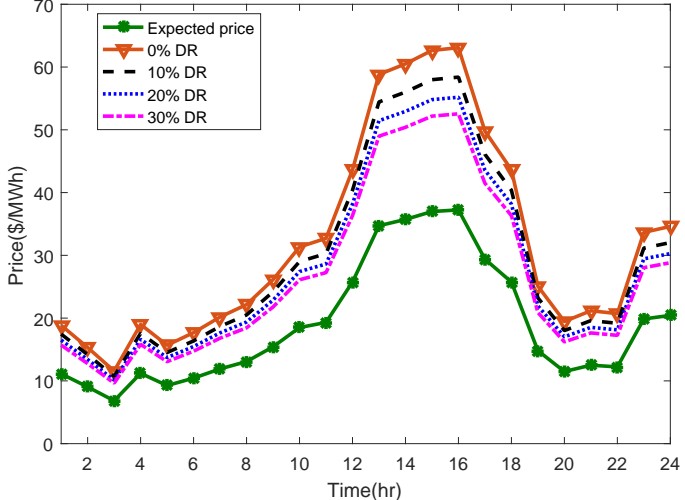

**Figure 9.** The effect of demand response on real-time prices.

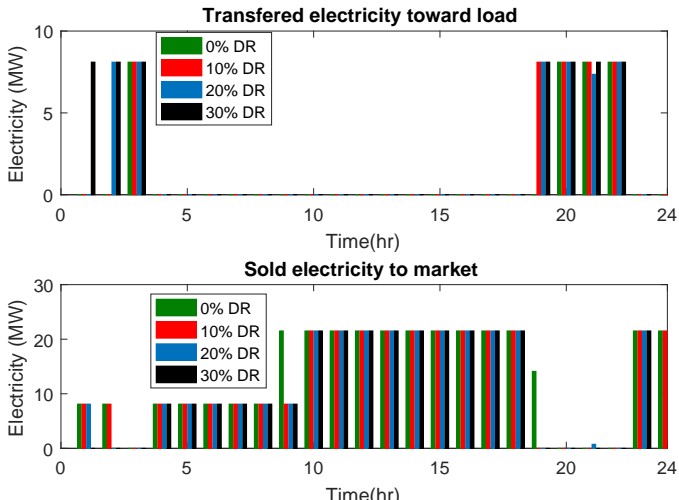

**Figure 10.** The power generated by the CHP.

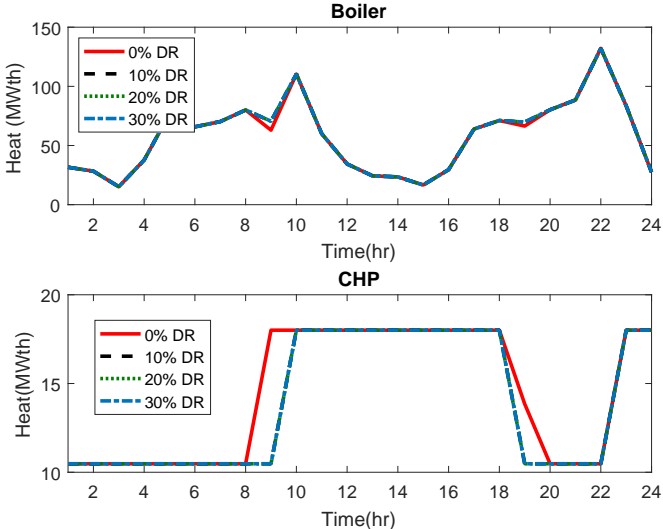

**Figure 11.** The heat generated by the boiler and the CHP.

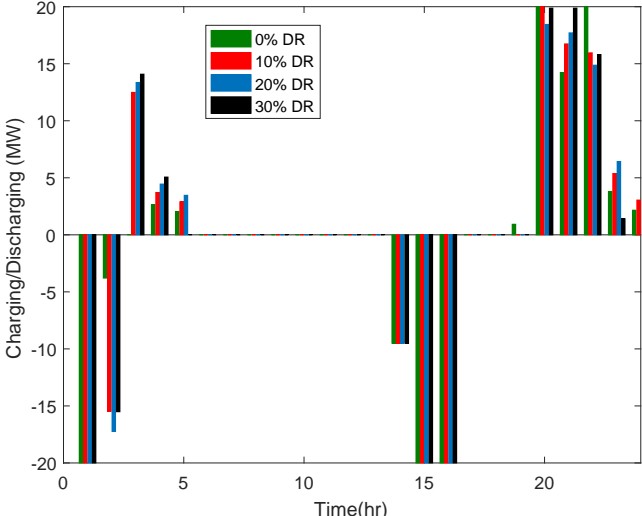

**Figure 12.** The charging/discharging schedule.

**Table 2.** A comparison of the stochastic programming method and Information Gap Decision Theory (IGDT).

| Percent of DR (%) | 0 | 10 | 20 | 30 |
|---|---|---|---|---|
| $\alpha$ | 0.694 | 0.568 | 0.482 | 0.411 |
| Expected cost ($) | 203,936 | 197,738 | 191,525 | 185,331 |
| Critical cost ($)($\Delta$) | 224,330 | 217,512 | 210,677 | 203,864 |
| Stochastic programming cost ($) | 201,342 | 194,737 | 188,132 | 181,528 |

**Table 3.** A sensitivity analysis on $\beta$ and the DR percentages.

| | $\beta$ | 0.02 | 0.04 | 0.06 | 0.08 | 0.1 | 0.12 |
|---|---|---|---|---|---|---|---|
| | $\alpha$ | 0.1 | 0.189 | 0.308 | 0.468 | 0.694 | 1 |
| | $P^{rt}(t)$ | 4348 | 3054 | 2535 | 1948 | 1333 | 1113 |
| 0 % of DR | $P^{DA}(t)$ | 4073 | 5439 | 6002 | 6638 | 7338 | 7638 |
| | $H^{CHP}(t)$ | 304 | 304 | 323 | 334 | 345 | 352 |
| | $P^{CHP}(t)$ | 288 | 288 | 322 | 341 | 361 | 374 |
| | $\beta$ | 0.02 | 0.04 | 0.06 | 0.08 | 0.1 | 0.12 |
| | $\alpha$ | 0.081 | 0.14 | 0.214 | 0.3 | 0.448 | 0.559 |
| | $P^{rt}(t)$ | 5227 | 4670 | 4496 | 3510 | 2675 | 1995 |
| 30 % of DR | $P^{DA}(t)$ | 3178 | 3769 | 3989 | 5016 | 5885 | 6628 |
| | $H^{CHP}(t)$ | 304 | 304 | 306 | 322 | 333 | 334 |
| | $P^{CHP}(t)$ | 288 | 288 | 292 | 320 | 342 | 342 |

## 7. Conclusions

This paper has proposed an IGDT-based model for solving the EH management problem by considering a DR program for the problem faced by large consumers of power systems to procure their energies. The EH has RT and DA electricity market prices, wind turbine production, and natural gas as inputs, while it has heat and electricity as outputs. The problem is modeled with IGDT, and the DR program is added to give large consumers more flexibility. The results demonstrate that, by increasing the percentage of DR, the peak load decreases and the off-peak demand increases. It can be seen that the uncertainty radius is reduced by increasing the percentage of DR. A high level of DR results in a smaller uncertainty radius and, consequently, lower RT prices and a higher value of wind turbine generation. By raising the risk coefficient, the level of participation in the RT and DA markets is dramatically reduced and increased, respectively. The CHP is affected by changing the risk coefficient in such a way that the higher the risk parameter, the lower the CHP generation.

**Author Contributions:** A.N.: conceptualization, methodology and writing original draft; M.M.: methodology and writing review; B.M.-I.: conceptualization, methodology and writing review; J.C.: conceptualization and writing review; M.P.-K. and M.L.: editing and data analysis; R.G.: editing revision version.

**Funding:** Radu Godina was supported by the Fundação para a Ciência e Tecnologia (grant UID/EMS/00667/2019). Javier Contreras was supported by the Ministry of Economy and Competitiveness of Spain under Project ENE2015-63879-R (MINECO/FEDER, UE).

**Acknowledgments:** Radu Godina would like to acknowledge financial support from Fundação para a Ciência e Tecnologia (grant UID/EMS/00667/2019).

**Conflicts of Interest:** The authors declare no conflict of interest.

## Nomenclature

**Indices**

| | |
|---|---|
| $t$ | Time index |
| $i$ | CHP index |
| $j$ | Boiler index |

**Variables**

| | |
|---|---|
| $C^{rt}/C^{da}$ | Total cost of purchasing from the RT/DA markets ($) |
| $P^{RT}/P^{DA}$ | Energy purchased from the RT/DA markets (MWh) |
| $SOC(t)$ | State of charge (SOC) (MWh) |
| $P^{dch}(t)/P^{ch}(t)$ | Discharging/charging power of the energy storage system (ESS) (MW) |
| $P_i^{CHP}(t)/H_i^{CHP}(t)$ | Electric/heat power produced by the combined heat and power (CHP) (MW) |
| $P_i^{CHP,M}(t)$ | Electric power sold to the market by the combined heat and power (CHP) (MW) |
| $P_i^{CHP,L}(t)$ | Electric power transferred to the load directly by the combined heat and power (CHP) (MW) |
| $C_{CHP}$ | Total cost of the CHP ($) |
| $H_j^{Blr}(t)$ | Heat power produced by the boiler (MWth) |
| $C_{Blr}$ | Total cost of the boiler ($) |
| $P_{CHP}^{gas}(t)/P_{Blr}^{gas}$ | Equivalent power of gas entering the CHP/ boiler (MW) |
| $D_e(t)$ | Electric power demand after the DR program |
| $DR(t)$ | Deployed DR |
| $ldr(t)$ | Shifted load in the DR program |
| $inc(t)$ | Increasing electricity demand in the DR program |
| $\alpha$ | Uncertainty radius |
| $\rho$ | Uncertain parameter of the IGDT problem |
| $f_b$ | Cost obtained with deterministic parameters |
| $\overline{P}^{WT}$ | Forecasted electric power produced by a wind turbine (MW) |
| $P_{LD}^{rt}(t)$ | Power purchased from the RT market to supply the load directly |
| $P_{LD}^{WT}(t)$ | Electric power generated by the wind turbine to supply the load directly |
| $P_{ST}^{rt}(t)$ | Power purchased from the RT market for charging the ESS |
| $P_{ST}^{WT}(t)$ | Electric power generated by the wind turbine for charging the ESS |
| $v^{CHP}(t)$ | Binary variable associated with the ON/OFF state of the CHP |
| $\zeta_{ch}/\zeta_{dch}$ | Binary variable associated with the ON/OFF state of charging/discharging |
| $u_{dr}(t)$ | Binary variable associated with decreasing electricity demand in the DR program |
| $u_{inc}(t)$ | Binary variable associated with increasing electricity demand in the DR program |

**Parameters**

| | |
|---|---|
| $P_{h,max}^{Blr}$ | Maximum generation of the boiler |
| $\eta_{Ch}^{ST}/\eta_{Dch}^{ST}$ | Charging/discharging efficiency |
| $E_{max}^{st}/E_{min}^{st}$ | Maximum/minimum capacity of the ESS |
| $E_{max}^{Dch}$ | Maximum discharge of the ESS |
| $\lambda^{DA}(t)$ | DA electricity market price ($/MWh) |
| $\overline{\lambda}^{RT}$ | Expected RT market price ($/MWh) |
| $inc^{max}$ | Maximum amount of increase in load in the DR program |
| $DR^{max}$ | Maximum amount of reduced demand in the DR program |
| $S_i$ | Wind speed (m/s) |
| $V_n$ | Nominal wind speed (m/s) |
| $V_{cin}/V_{cout}$ | Cut-in/cut out wind speed (m/s) |
| $P_n$ | Nominal power of a wind turbine (MW) |
| $D_e^0(t)$ | Base electricity demand |
| $D_h(t)$ | Heat demand |
| $\beta$ | Risk parameter |
| $\Delta$ | Critical value of the objective function |
| $\overline{\rho}$ | Expected value of an uncertain parameter for IGDT modeling |
| $\lambda^{gas}$ | Gas price ($) |
| $\overline{P}_{LD}^{rt}(t)$ | Expected value of the power purchased from the RT market to supply the load directly |

| | |
|---|---|
| $\overline{P}_{\text{LD}}^{WT}(t)$ | Expected value of the electric power generated by a wind turbine to supply the load directly |
| $\overline{P}_{\text{ST}}^{\text{rt}}$ | Expected value of the power purchased from the RT market for charging the ESS |
| $\overline{P}_{\text{ST}}^{WT}$ | Expected value of the electric power generated by the wind turbine for charging the ESS |
| $a_i, b_i, c_i, d_i, e_i, f_i,$ | Cost coefficients of the CHP |
| **Sets** | |
| $T$ | Number of time periods |
| $I$ | Number of CHP units |
| $J$ | Nofumer of boiler units |

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
