# Peer review of "Uncertainty-Based Models for Optimal Management of Energy Hubs Considering Demand Response"

_energies, doi:10.3390/en12081413_

Round 1

Reviewer 1 Report

The information gap decision theory considering DR in a energy hub model was discussed in this work. This reviewer would have the following comments for the author to address: 

 Abstract, please summarize what time frame energy hub problem are you addressing in this work, operational, planning?

Information gas decision theory, page 9, what type of problem is this method widely used before you adopt this for solving the proposed model? Pleas add some background information to discuss, this reviewer is expecting some similarities between those problems and the energy hub mode in this work so that method also apply. 

Page 11, 5.2, here it would be helpful to list all the uncertainties factors or variables when you present the formulation. 

The color yellow used in the figure might need to be changed to adapt black/white background reading. 

Reference on energy hub topic with energy storage in operational time frame, with transmission in planning horizon may be discussed to enhance the literature review. 

"On decisive storage parameters for minimizing energy supply costs in multicarrier energy systems", IEEE Trans. Sustain. Energy, vol. 5, no. 1, pp. 102-109, Jan. 2014.

Optimal Expansion Planning of Energy Hub With Multiple Energy Infrastructures IEEE Transactions on Smart Grid, Volume: 6 , Issue: 5 , Sept. 2015

Round 2

Reviewer 2 Report

The new copy of paper is well written and discussed an interesting topic.  Authors replied to all comments and no further comments from the reviewer.